# Study on the Mechanical Properties of Cast-In-Situ Phosphogypsum as Building Material for Structural Walls

**DOI:** 10.3390/ma16041481

**Published:** 2023-02-10

**Authors:** Qizhu Yang, Ze Xiang, Taoyong Liu, Changqing Deng, Huagang Zhang

**Affiliations:** 1College of Civil Engineering and Architectural, Shaoyang University, Shaoyang 422000, China; 2Space Structures Research Center, Guizhou University, Guiyang 550025, China

**Keywords:** cast-in-situ phosphogypsum, stress–strain curve, elastic modulus, Poisson’s ratio

## Abstract

The application of cast-in-situ phosphogypsum as the wall material of building structures can greatly reduce the environmental pollution caused by phosphogypsum. Through the uniaxial compression test of cast-in-situ phosphogypsum specimens, the compressive strength of cast-in-situ phosphogypsum is determined, the constitutive relationship of the material is drawn up, and the elastic modulus and Poisson’s ratio of the material are determined. The results show that when the strain of the specimen is close to the peak strain, the cast-in-situ phosphogypsum has brittle properties and rapidly fails, where the failure state is mainly splitting failure. The retarder has a great influence on the peak stress. When the content of the retarder is about 0.3%, the peak stress is 8.6 MPa and the ultimate strain is 2.54 × 10^−3^, while the peak stress is 2.8 MPa and the ultimate strain is 2.01 × 10^−3^. The three segment constitutive fitted equations reflect all the characteristics of the compression specimen. When the strength of the cast-in-situ phosphogypsum is high, the elastic modulus is also high. When the content of the retarder is about 0.3%, the elastic modulus is 5300 MPa, and when the content of retarder is far greater than 0.3%, the elastic modulus is 2000 MPa. The Poisson’s ratio of material is recommended as 0.19.

## 1. Introduction

With the rapid development of the economy, developments in architecture are also occurring with each passing day. On the one hand, a large number of new structural forms have emerged from the innovation of building structures [1]; on the other hand, breakthroughs have been made in building materials, especially in the research of composite materials, reinforced polymers and other aspects, and certain achievements have been made [2,3]. The technology of 3D printing has also been applied [4], and the utilization of phosphogypsum resources has been involved in the whole process [5]. China’s phosphate ore resources are mainly distributed in Hubei, Yunnan, Guizhou, Sichuan, Hunan and five other provinces, and it is one of the largest phosphate producing countries in the world [6]. Phosphorous chemical products are widely used in agriculture, the chemical industry, medicine and other fields [7], play an important role in the development of the national economy, and also produce a large amount of phosphogypsum. The vast majority of this phosphogypsum is stacked as waste. The investment needed for storage yards is large, the operation costs are high and they occupy a lot of land resources. The environmental pollution caused by the infiltration of harmful substances, such as soluble P_2_O_5_ and fluoride, in phosphogypsum into the soil and water systems is also increasingly prominent [8,9]. However, the comprehensive utilization rate of phosphogypsum is low in China; it is mainly used to produce building materials with low added value. In this regard, the research and development of new technologies and new products of phosphogypsum should be accelerated. If industrial waste gypsum can be modified and comprehensively utilized, it can be turned into treasure. At present, the comprehensive utilization of phosphogypsum is mainly reflected in the following aspects.

First, in the development of new chemical products after reprocessing. For example, in the literature [10], CaO-Al2O3-SiO2 glass-ceramic was prepared from phosphogypsum and studied by differential thermal analysis (DTA) and X-ray diffraction (XRD). The work [11] describes a method of successfully preparing high-quality nano-calcium fluoride in aqueous solutions by direct precipitation using calcined phosphogypsum as a raw material. The work [12] introduces a continuous and large-scale method for manufacturing phosphogypsum composites through a reactive extrusion strategy. The work [13] proposed a simple and efficient approach for the high-purity CaSO_4_·2H_2_O (DH) whiskers and α-CaSO_4_·0.5H_2_O (α-HH) whiskers derived from phosphogypsum (PG). The work [14] studied the effect of different conditions on the preparation of CaS by reducing PG with coal in a fluidized bed. This method not only requires a large initial investment, but also forms new pollutants and faces environmental pollution problems. 

Second, in its use as a soil conditioner, where a large number of scholars have carried out relevant research. The literature [15] describes how to industrially process phosphogypsum into organic mineral fertilizer. The literature [16] studied the application of lime and phosphogypsum in weathered soil and determined the long-term (10 year) impact of lime and phosphogypsum on the soil surface of tropical NT soil C and N components; however, the application rate was low. 

The third is in the processing of building and decoration materials with phosphogypsum, such as cement retarder and block, gypsum board and wire. The influence of phosphogypsum on cement setting and its applications are described in the literature [17,18,19] over many years, which shows that the preparation of cement retarders is one of the main ways for the resource utilization of phosphogypsum. The works [20,21,22] introduced the early-stage development of several phosphogypsum blocks and the current applications of unburned phosphogypsum permeable bricks. The works [23,24,25,26] introduced new technologies related to the preparation of phosphogypsum rapid wallboard and cast-in-situ phosphogypsum walls. Their application could consume a lot of pollutants and improve construction efficiency. The work [27] compared and analyzed the chemical composition and pretreatment methods of different phosphogypsum, and summarized the use of phosphogypsum in cement retarders, building materials, cement raw materials, concrete raw materials, cementitious materials, fillers and modifiers. However, the consumption of phosphogypsum in these materials is small, which does not meet expectations, and the construction speed of block masonry is also slow. 

Fourth, is its use in subgrade engineering and mine backfilling [28,29]. Measures should be taken to solidify phosphogypsum and isolate it from soil to prevent leakage pollution. However, the demand for gypsum in the above fields is limited, and China is rich in natural gypsum reserves, which makes it difficult to achieve the purpose of processing industrial waste. Only by modifying industrial waste phosphogypsum, transforming it into main building materials and expanding its application space, can we expect to achieve the purpose of large-scale treatment of industrial waste.

The literature [7] summarizes the research progress of phosphogypsum in the fields of construction, rare earth extraction, agriculture, chemical industry and biomedical industry; outlines the current problems in the comprehensive utilization research and industrial application of phosphogypsum; and looks forward to the developing trend of basic theory and technology research of high value processing of phosphogypsum in the future. In order to maximize the use of phosphogypsum, turn waste into treasure and save resources, this work uses cast-in-situ phosphogypsum as the building material for structure walls, which can greatly reduce the environmental pollution by phosphogypsum. The strength of phosphogypsum material is an important mechanical parameter in its engineering application and mechanical properties are the main basis for studying phosphogypsum as a building material for structural walls. Therefore, this paper discusses the engineering mix ratio of the cast-in-situ phosphogypsum material, synthesizes test pieces to measure the engineering application technical performance indicators, such as the setting time, moisture content and radioactivity of the cast-in-situ phosphogypsum, and opens the research on the mechanical properties of the cast-in-situ phosphogypsum material. Through compressive tests on phosphogypsum cube and prism specimens, the compressive strength of cast-in-situ phosphogypsum is measured, the partial coefficient of the materials is given, the constitutive relationship of materials is drawn up and the mechanical performance indexes, such as the elastic modulus and Poisson’s ratio, of materials are measured for reference in engineering applications.

## 2. Materials and Methods

### 2.1. Raw Materials

The raw materials of cast-in-situ phosphogypsum are mainly phosphogypsum, phosphorus slag powder, hydrated lime, cement, a retarder and a water reducer. Through preliminary investigations, it was found that the gypsum phase of phosphogypsum produced by different phosphoric acid plants is slightly different. Therefore, the gypsum phase of phosphogypsum raw materials should be tested before application. Figure 1 shows the construction photos of cast-in-situ phosphogypsum walls and treated phosphogypsum. The phosphogypsum and phosphorous slag micropowder were taken from Guizhou Wengfu Phosphorus Industry Group. The gypsum phase composition of phosphogypsum was determined and is shown in Table 1. The specific surface area of the phosphorus slag powder was 380~420 m^2^/kg and its chemical composition is shown in Table 2. The hydrated lime is commercially available, and the mass fraction of effective CaO was no less than 60%. The cement is P.O325 ordinary Portland cement and the water-reducing agent was polycarboxylic acid, with a concentration of no less than 10%. The retarder was sodium citrate. A JMS-6490LV scanning electron microscope was used to analyze the morphology of the phosphogypsum and phosphorus slag powder. The results are shown in Figure 2, where the scale bar in Figure 2a is 100 μm, and the scale bar in Figure 2b is 10 μm. The crystals of hemihydrate gypsum in phosphogypsum are generally parallelogram plates, and the gap between crystals is large. The phosphorus slag micro powder particles are in the shape of “gravel”, with clear edges and corners but no fixed cleavage surface. The results of the X-ray diffraction (SIEMENS D5000 XRD diffractometer, Munich, Germany ) analysis are shown in Figure 3. The phosphorus slag powder is mainly composed of glass, containing a small amount of pseudowollastonite, gunite and apatite. It is a volcanic material with potential activity and can be filled in the gap between hemihydrate gypsum crystals after casting. The acidity of phosphogypsum is neutralized by hydrated lime, the anhydrite is consumed by cement, the water reducer is used to reduce the water consumption, and the retarder is used to delay the setting time after being cast-in-situ.

Taking the total mass mix proportion of phosphogypsum, phosphorous slag micropowder and hydrated lime as 100%, the cement consumption was calculated according to the mass of the mixture of phosphogypsum, phosphorous slag micropowder and hydrated lime. The mixed dry materials were obtained after uniform mixing. The consumption of water reducer and retarder was calculated according to the mass of the mixed dry materials. Considering the possible construction deviation, the material mix proportion shown in Table 3 is proposed, and the water cement ratio is 0.43.

### 2.2. Technical Parameters of Cast-In-Situ Phosphogypsum

Referring to the Chinese Standard of Building Gypsum GB/T9776 and the Standard of Radionuclide Limit of Building Materials GB6566, 40 mm × 40 mm × 160 mm cast-in-situ phosphogypsum specimens were synthesized and the technical performance indexes such as setting time, water content and radioactivity were evaluated.

#### 2.2.1. Setting Time

Table 4 shows the setting time test results of the phosphogypsum raw materials and cast-in-situ phosphogypsum, in which the amount of retarder added to cast-in-situ phosphogypsum is 0.3%. Compared with the phosphogypsum raw materials, the initial setting time of cast-in-situ phosphogypsum is 7.5 times more than the initial setting time of the raw materials, and the final setting time is about 8 times more than the final setting time of raw materials. The setting time is more than 30 min, which is conducive to on-site pouring.

#### 2.2.2. Moisture Content

The moisture content of cast-in-situ phosphogypsum measured based on 0.3% retarder is shown in Table 5. The moisture content after 5 days at 10 ± 7 °C is 2.6%, so the application of cast-in-situ phosphogypsum to the wall will not affect the building decoration.

#### 2.2.3. Radioactive Detection

The radioactive detection results of cast-in-situ phosphogypsum are shown in Table 6. The radioactive indicators meet the radioactive requirements for main building materials in the Limits of Radionuclides in Building Materials (GB6566). Therefore, the cast-in-situ phosphogypsum can be used for residential buildings.

### 2.3. Test Device and Loading System

Figure 4 shows a self-made loading device [30]. Four springs with the same stiffness were used as buffer elements. The loading stiffness was 89.7 kN/cm and the unloading stiffness was 92.0 kN/cm. The spring has manufacturing and installation errors; therefore, the load transmission of the loading device has eccentric effects, but it can be considered as a systematic error.

The load sensor is welded by two 63 mm × 5 mm angled steel tips to measure the pressure on the test piece. The total load sensor was made by butt welding the channel steel tips to control the loading speed. The load was continuously applied by a 50 t hydraulic jack, and the loading speed was controlled at 0.5~1.5 MPa/s.

In order to minimize the impact of the loading device on the measurement accuracy, we used a 3 mm-thick steel plate to make two steel hoops to clamp the test pieces up and down, and then installed a longitudinal displacement meter, as shown in Figure 4c. The lateral displacement meter contacts the plexiglass placed on the test piece through a jackscrew installed on the suspension plexiglass sheet. Corresponding to the mounting surface of the longitudinal displacement meter, the longitudinal and transverse strain gauges were, respectively, placed in the middle of the specimen height direction to collect the strain value before the specimen cracks. Two longitudinal and transverse displacement gauges and four longitudinal and transverse strain gauges were used. The arrangement of measuring points is shown in Figure 4c, and the installation of test pieces is shown in Figure 4e.

## 3. Results

### 3.1. Test Phenomenon

A total of 18 cube specimens and 38 prism specimens were synthesized according to the mix proportion in Table 3, and an axial compression test was carried out for each. The theoretical side length of the cube specimens was 100 mm, and the theoretical dimensions of the prism specimens was 100 mm × 100 mm × 300 mm. The loading process can be roughly divided into three stages. When the load is 0.13~0.2 times the peak load, the cast-in-situ phosphogypsum between the upper loading plate and the steel hoop will fall off on the surface, indicating that the specimen is in the compaction stage. When the load reached 0.8~0.9 times the peak load, cracks appear on the surface of the specimen. With continued loading, the cracks extended to the upper and lower ends of the test piece and quickly penetrated the test piece, destroying it. The destruction process was relatively sudden and short, indicating that the cast-in-situ phosphogypsum has a high brittleness.

The typical failure state of the cube specimen is shown in Figure 5. The failure state can be generally divided into cone failure and split failure. The split failure surface is mainly in the vertical and 45-degree planes, as shown in Figure 5a. The conical failure state is shown in Figure 5b, and there is no necessary relationship between the failure state and the material mix proportion. The failure state of prismatic specimen is mainly splitting failure, as shown in Figure 6.

### 3.2. Volumetric Weight of Cast-In-Situ Phosphogypsum

The volumetric weight of the cast-in-situ phosphogypsum is mainly affected by the phosphogypsum and phosphorus slag powder. Through a statistical analysis of the test results of the volumetric weight of the test pieces, it is recommended that the volumetric weight of the cast-in-situ phosphogypsum be 14.0 kN/m^3^–15.0 kN/m^3^ when using the material mix ratio of this project.

### 3.3. Compressive Strength and Material Partial Factor

According to axial compressive strength tests of the cube and prism specimens, the compressive strength and its statistical parameters of phosphogypsum are shown in Table 7.

In order to analyze the reliability of the bearing capacity of cast-in-situ phosphogypsum walls, an axial compression test of a 1/2 scale wall model was carried out. The section size of the specimen was a × b = 200 mm × 120 mm, 450 mm, 750 mm and 1050 mm in height. See Figure 7 for the test results.

Based on the bearing capacity test results of the wall model and the actual engineering application needs and after the reliability analysis of the bearing capacity of the cast-in-situ phosphogypsum wall, it is recommended that the material sub coefficient of the cast-in-situ phosphogypsum be taken as γ_P_ = 1.9. The standard value and design value of compressive strength of each mix proportion material are shown in Table 8.

### 3.4. Analysis of Strength Obtaining Mechanism of Cast-in-situ Phosphogypsum

The strength of the cast-in-situ phosphogypsum paste was obtained during the material hardening process. After the setting crystals interlace, the friction and cohesion between crystals strengthen the material paste. When setting and hardening stop, the strength will not increase. Phosphorus slag micropowder is a kind of potentially active hydraulic cementing material. Adding phosphorus slag micropowder can delay the hydration time of hemihydrate gypsum in the slurry, increase the fluidity of the cast-in-situ slurry and fill in the gap between dihydrate gypsum crystals to improve the water resistance of cast-in-situ phosphogypsum. The purpose of adding a retarder is to reduce the solubility and dissolution rate of hemihydrate calcium sulfate in phosphogypsum, delay the reduction time of hemihydrate calcium sulfate and thus prolong the setting time of the cast-in-situ phosphogypsum paste. The function of cement is to consume anhydrite in phosphogypsum.

The results show that the retarder has a great influence on the compressive strength of the cast-in-situ phosphogypsum, and it is suggested that the dose of retarder should be controlled between 0.2% and 0.4%. When the content of anhydrous calcium sulfate in phosphogypsum is high, the mix proportion of cement can be increased accordingly. When the mix proportion of retarder is about 0.3%, the quality mix proportion of cement can be controlled between 2.0% and 5.0%.

### 3.5. Material Constitutive Relationship of Cast-in-situ Phosphogypsum

In view of the high brittleness of the cast-in-situ phosphogypsum, according to the axial compression test results of the prism specimen with a side length of 100 mm × 100 mm × 300 mm, a three-stage stress–strain relationship curve of the cast-in-situ phosphogypsum material is proposed using dimensionless coordinates *x* = *ε*/*ε*_0_ and *y* = *σ*/*σ*_0_, as shown in Figure 8.

In Figure 8, *σ* and *ε* are the stress and strain of the cast-in-situ phosphogypsum, respectively, *σ*_0_ is the peak stress, *ε*_0_ is the peak strain corresponding to *σ*_0_, *σ*_r_ is the residual stress, *ε*_u_ is the limit strain corresponding to *σ*_r_. The statistical control parameters of the constitutive curve are shown in Table 9.

The curve equation can be divided into:①When the dosage of retarder is about 0.3%,
(1)$$y\left(x\right)=\{\begin{array}{l} 0.395x+1.675x^{2}\;\;\;\;\;\;\;\left(0\leq x\leq 0.2\right)\\ 0.328x+2.344x^{2}-1.672x^{3}\;\left(0.2\leq x\leq 1.0\right)\\ 1-1.008{\left(x-1\right)}^{2}\;\;\;\;\;\;\;\;\;\left(x\geq 1.0\right) \end{array}.$$

②When the dosage of retarder is far greater than 0.3%,


(2)
$$y\left(x\right)=\{\begin{array}{l} 0.378x+1.704x^{2}\;\;\;\;\;\;\;\;\left(0\leq x\leq 0.2\right)\\ 0.311x+2.38x^{2}-1.69x^{3}\;\;\;\left(0.2\leq x\leq 1.0\right)\\ 1-0.933{\left(x-1\right)}^{2}\;\;\;\;\;\;\;\;\;\;\left(x\geq 1.0\right) \end{array}.$$


Since the inflection point coordinates of the curves in Formulas (1) and (2) are relatively close, the recommended constitutive relationship of cast-in-situ phosphogypsum in this project is:(3)$$y\left(x\right)=\{\begin{array}{l} 0.386x+1.689x^{2}\;\;\;\;\;\;\;\left(0\leq x\leq 0.2\right)\\ 0.319x+2.362x^{2}-1.681x^{3}\;\left(0.2\leq x\leq 1.0\right)\\ 1-0.975{\left(x-1\right)}^{2}\;\;\;\;\;\;\;\;\;\left(x\geq 1.0\right) \end{array}.$$

At this time, ε_u_/ε_0_ = 1.45 and σ_r_/σ_0_ = 0.7.

### 3.6. Elastic Modulus and Poisson’s Ratio

Due to the high brittleness of the cast-in-situ phosphogypsum, there is almost no linear phase in the constitutive relationship, so the elastic modulus of materials in this project was comprehensively evaluated according to the following three conditions:The average value, *E*_*t*1_, of the secant elastic modulus was calculated based on the stress–strain test results when the strain values are 0.2*ε*_0_ and 0.45*ε*_0_;The measured results of stress–strain at *σ* = 0.4*σ*_0_ were used to calculate the average value *E*_*t*2_ of secant elastic modulus across the origin;According to the test results, under the condition of no lateral pressure, when $\varepsilon \leq 0.2\varepsilon_{0}$, the cast-in-situ phosphogypsum is under load at the compaction stage, and the material constitutive relationship curve has an inflection point between $0.2\varepsilon_{0}$ and $\varepsilon_{0}$. Therefore, the inflection point was moved to *x* = 0.45, linear regression was conducted on the measured stress–strain results between $0.2\varepsilon_{0}$ and $0.45\varepsilon_{0}$ and the average slope of the regression line was approximated as the material elastic modulus, *E*_*t*3_.

The relationship between the three elastic moduli is shown in Figure 9, where *α*_1_, *α*_2_ and *α*_3_ correspond to *E*_*t*1_, *E*_*t*2_ and *E*_*t*3_, respectively. For safety, the material elastic modulus of the cast-in-situ phosphogypsum for each mix proportion is given according to the secant elastic modulus, *E*_*t*2_, at the origin, as shown in Table 10. When the content of the retarder is about 0.3%, the elastic modulus is *E* = 5300 N/mm^2^ and the variation coefficient is *δ_E_* = 0.172. When the content of the retarder is much more than 0.3%, the elastic modulus is *E* = 2000 N/mm^2^ and the variation coefficient is *δ_E_* = 0.324.

The Poisson’s ratio was calculated from the previous longitudinal and transverse strain test values when *σ* = 0.4*σ*_0_, and the strain test results when the Poisson’s ratio tends to be stable were used for linear regression. The Poisson’s ratio results of materials for each mix ratio are shown in Table 10. The value of *ν* changes slightly. All test results were statistically analyzed to obtain a Poisson’s ratio of *ν* = 0.19 and a coefficient of variation of *δ_ν_* = 0.087.

## 4. Conclusions

Using cast-in-situ phosphogypsum as a wall material for buildings can consume a large amount of pollutants and improve the efficiency of building construction. The strength of the phosphogypsum material is an important mechanical parameter in engineering applications, and its mechanical properties are the main basis for studying phosphogypsum as a wall material. Through completing relevant tests on cast-in-situ phosphogypsum specimens, the results show that:(1)The failure of cast-in-situ phosphogypsum uniaxial compression specimens is brittle failure, manifesting in a splitting failure state. The main cracks include both vertical cracks and oblique cracks.(2)The descending section of the measured stress strain curve of cast-in-situ phosphogypsum is relatively short, and the amount of retarder added has a great impact on the strength of the material. Generally, when the amount of retarder added is about 0.3%, the peak stress is 8.6 MPa and the limit strain is 2.54 × 10^−3^. When the content of retarder is much more than 0.3%, the peak stress is 2.8 MPa and the limit strain is 2.01 × 10^−3^.(3)The constitutive relation curve of cast-in-situ phosphogypsum is composed of an ascending section and a descending section. The Formula (2) proposed in this paper fits the test results of ascending section well, but the fit of the descending section is poor.(4)In general, when the content of retarder is about 0.3%, the elastic modulus of cast-in-situ phosphogypsum is *E* = 5300 MPa, and when the content of retarder is far greater than 0.3%, the elastic modulus is *E* = 2000 MPa.(5)The test results show that there is no necessary relationship between the Poisson’s ratio and the strength of the cast-in-situ phosphogypsum. It is suggested that the Poisson’s ratio should be 0.19.

## Figures and Tables

**Figure 1 materials-16-01481-f001:**
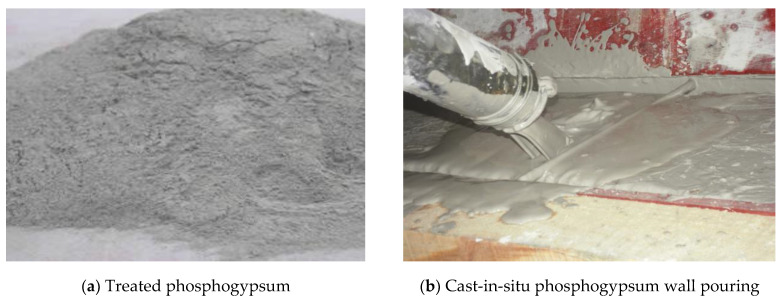
Cast-in-situ phosphogypsum wall pouring and phosphogypsum samples.

**Figure 2 materials-16-01481-f002:**
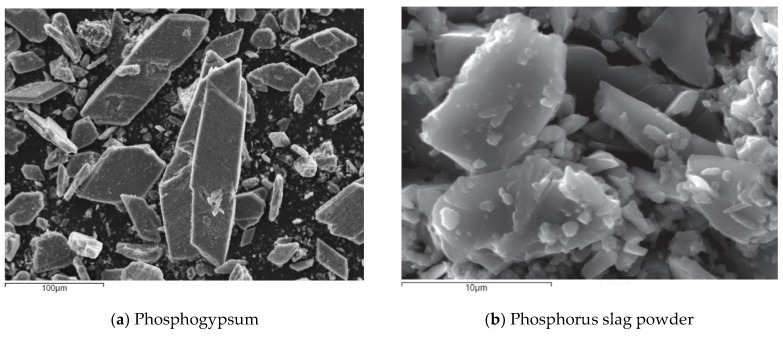
SME photos of raw materials.

**Figure 3 materials-16-01481-f003:**
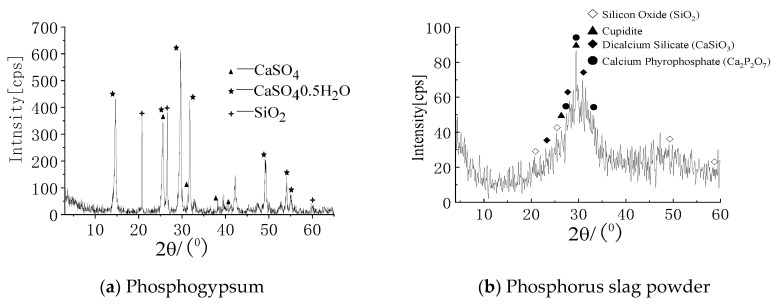
XRD atlas of raw materials.

**Figure 4 materials-16-01481-f004:**
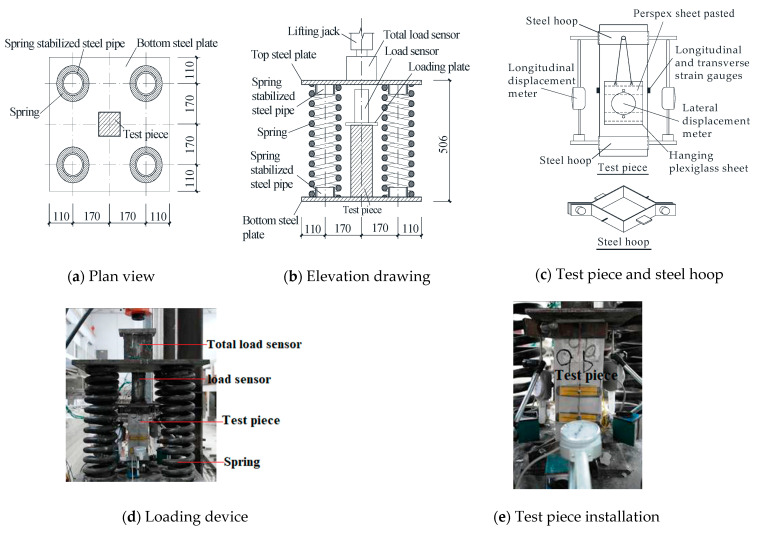
Test equipment and measuring point arrangement.

**Figure 5 materials-16-01481-f005:**
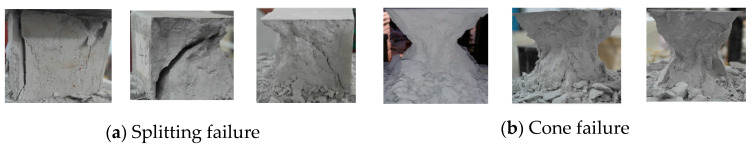
Typical failure state of cube specimen.

**Figure 6 materials-16-01481-f006:**
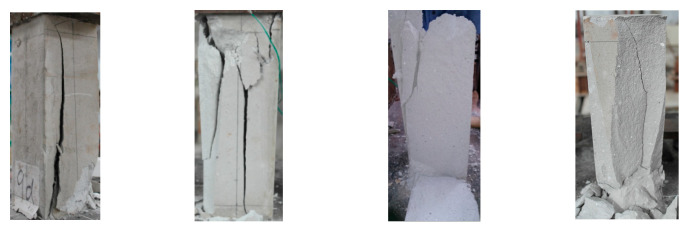
Typical failure state of prism specimen.

**Figure 7 materials-16-01481-f007:**
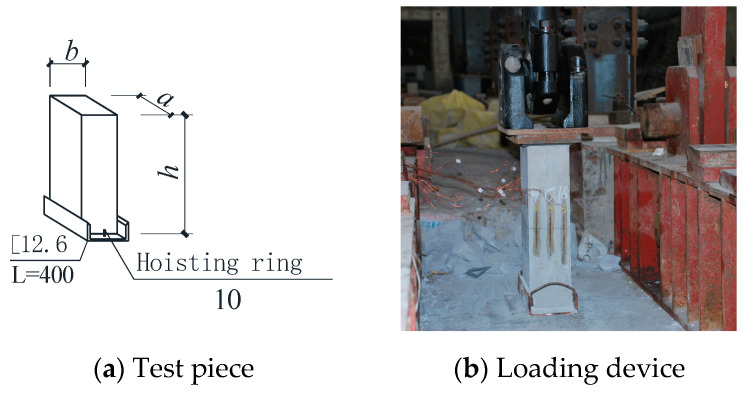
Wall axial compression test.

**Figure 8 materials-16-01481-f008:**
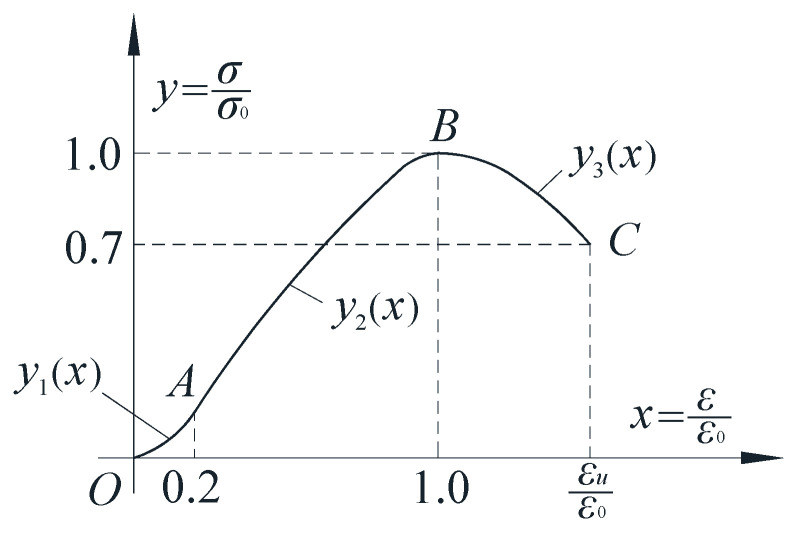
Fitting curve of constitutive relation.

**Figure 9 materials-16-01481-f009:**
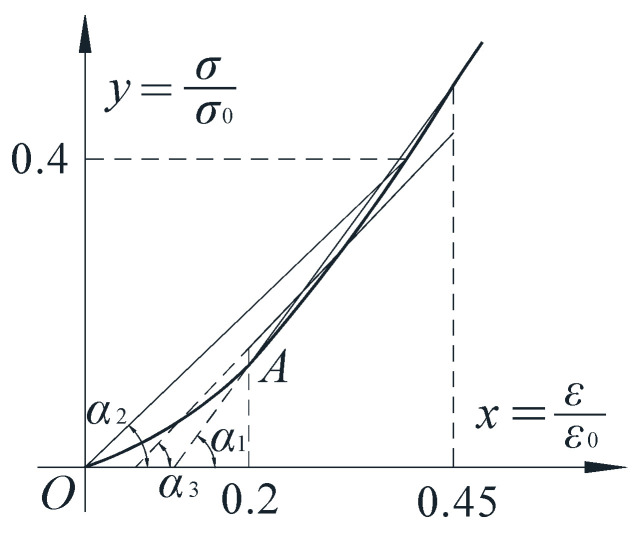
Comparison of elastic modulus.

**Table 1 materials-16-01481-t001:** Gypsum composition of phosphogypsum.

Dihydrate Gypsum/%	β-Hemihydrate/%	Anhydrite/%	
		Solubility	Difficult Capacitive
0.78	87.61	2.45	2.12

**Table 2 materials-16-01481-t002:** Chemical compositions of phosphorite slag (%).

SiO_2_	Fe_2_O_3_	Al_2_O_3_	CaO	MgO	P_2_O_5_	CaF_2_
42.35	1.67	3.52	45.36	1.73	1.46	1.07

**Table 3 materials-16-01481-t003:** Mass proportion of materials of specimens.

Material Composition/%	Mix Ratio Number								
	1	2	3	4	5	6	7	8	9
Phosphogypsum	75	75	75	80	80	80	85	85	85
Phosphorus slag powder	20	20	20	17	17	17	9	9	9
Slaked lime	5	5	5	3	3	3	6	6	6
Cement	5	2	10	5	2	10	5	2	10
Water reducing admixture	0.6	0.8	1.0	0.8	1.0	0.6	1.0	0.6	0.8
Retarder	1.0	0.3	1.5	1.5	1.0	0.3	0.2–0.4	1.5	1.0

**Table 4 materials-16-01481-t004:** Setting time/min.

Specimen	Initial Set	Final Set
Phosphogypsum	6.0	7.0
Cast-in-situ phosphogypsum	45.0	55.0

**Table 5 materials-16-01481-t005:** Moisture content of cast-in-situ phosphogypsum.

Initial Moisture Content/%	1d Moisture Content/%	3d Moisture Content/%	5d Moisture Content/%	Apparent State
28.5	18.3	4.7	2.6	Smooth

**Table 6 materials-16-01481-t006:** Radioactive Detection of Cast-in-situ Phosphogypsum.

Project	Test Data and Standard Values				
	^226^Ra/Bq.kg^−1^	^232^Th/Bq.kg^−1^	^40^K/Bq.kg^−1^	Internal Exposure Index/I_Ra_	External Exposure Index/I_γ_
Phosphorus building gypsum wall grouting material	52.31	18.62	243.21	0.31	0.34
Technical requirements for main building materials	-	-	-	<1.0	<1.0

**Table 7 materials-16-01481-t007:** Compressive strength and its coefficient of variation.

Mix Ratio Number	Cube		Prismatic		*f*_pu,10_/*f*_pu_
	*f*_pu_/MPa	*δ* _fpu_	*f*_pu,10_/MPa	*δ* _fpu,10_	
1	3.3	0.082	2.8	0.112	0.848
2	9.6	0.099	8.7	0.107	0.906
3	2.7	0.098	-	-	-
4	3.1	0.066	2.4	0.442	0.774
5	4.4	0.081	2.6	0.282	0.591
6	10.4	0.051	9.3	0.173	0.894
7	10.3	0.072	8.5	0.052	0.825
8	2.7	0.071	2.1	0.043	0.778
9	4.2	0.103	3.5	0.121	0.833

**Table 8 materials-16-01481-t008:** Standard value and design value of material compressive strength.

Mix Ratio Number	1	2	3	4	5	6	7	8	9
*f*_pk_/MPa	1.8	5.3	1.5	1.7	2.4	5.7	5.7	1.5	2.3
*f*_p_/MPa	0.9	2.8	0.8	0.9	1.3	3.0	3.0	0.8	1.2

**Table 9 materials-16-01481-t009:** Statistics of the control parameters of the stress–strain curve.

Dosage of Retarder	*σ*_0_/MPa	*δ_σ_* _0_	*σ*_r_/MPa	*δ_σ_* _r_	*σ* _r_ */σ* _0_	*ε*_0_/10^−6^	*δ_ε_* _0_	*ε*_u_/10^−6^	*δ_ε_* _u_	*ε*_u_/*ε*_0_
Far more than 0.3%	2.8	0.235	1.9	0.304	0.68	1430	0.109	2010	0.158	1.41
About 0.3%	8.6	0.072	5.7	0.160	0.66	1690	0.093	2540	0.142	1.50

**Table 10 materials-16-01481-t010:** Elastic modulus and Poisson’s ratio of cast-in-situ phosphogypsum.

Mix Ratio Number	1	2	3	4	5	6	7	8	9
Elastic modulus *E*/N.mm^−2^	2200	5600	1100	2000	1600	5300	5200	1900	2000
Poisson’s ratio, *ν*	0.16	0.20	0.18	0.18	0.18	0.20	0.20	0.2	0.2

## Data Availability

Data sharing not applicable.

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
