# Peer review of "Study on the Mechanical Properties of Cast-In-Situ Phosphogypsum as Building Material for Structural Walls"

_materials, 2023, doi:10.3390/ma16041481_

Round 1

Reviewer 1 Report

The presented manuscript includes the study of friction stir alloying of study on mechanical properties of cast-in-situ phosphogypsum as wall material of building structure.

The paper is of interest. The results of the work are presented on a good level but some questions and weak points should be mentioned.

1. Scale bar have to be added to the SEM images. Fig.1.

2. All obtained results firstly have to be compared with the published analogues; secondly, they have to be compared with the standards. It could be done both in text and final table.

3. How many samples were tested in parallel? Please add standard deviation values where applicable

Reviewer 2 Report

The literature review is not complete. The added value of the work should be backed up with comparison against works such as:

- Li, Y., Dai, S., Zhang, Y., Huang, J., Su, Y., & Ma, B. (2018). Preparation and thermal insulation performance of cast-in-situ phosphogypsum wall. Journal of applied biomaterials & functional materials, 16(1_suppl), 81-92.

- Zhang, Y., Dai, S., Weng, W., Huang, J., Su, Y., & Cai, Y. (2017). Stress-strain relationship and seismic performance of cast-in-situ phosphogypsum. Journal of Applied Biomaterials & Functional Materials, 15(1_suppl), 62-68.

- Bandgar, G. S., & Kumthekar, M. B. (2016). A Study on Feasibility of Rapid Wall Panel for Building Construction. International Research Journal of Engineering and Technology, 3(6), 1026-1031.

Other types of building should be mentioned as well, enriching the introduciton: Muñoz, I., Alonso-Madrid, J., Menéndez-Muñiz, M., Uhart, M., Canou, J., Martin, C., ... & Stavropoulos, P. (2021). Life cycle assessment of integrated additive–subtractive concrete 3D printing. The International Journal of Advanced Manufacturing Technology, 112, 2149-2159.

For the method, it is suggested to add a drawing showing the measured values implicated in the estimation of (A) Young modulus and (B) Poisson ratio

Also, a comparison with other materials should be made for those two values.

Reviewer 3 Report

In this manuscript, the mechanical properties of cast-in-situ phosphogypsum as wall material of building structure were investigated experimentally. Some suggestions and questions are provided below to be applied and answered before accepting the manuscript:

-       What is the main reason behind selecting cast-in-situ in the title and the manuscript’s text?

-       In the section 2, it is suggested to add some figures from the material itself.

-       In Figure 3, instead of addressing different part as numbers, it is suggested to add the name of each part within the figure.

-       Figure 6 can be presented with larger size. Some fonts in this figure are illegible.

-       In Figures 7 and 8, please check the notes. There are illegible.

-       There are some other related publications which can be added to the literature review. Some of the related publications provided below:

·         Xiantao, Q., Yihu, C., Haowei, G., Qisheng, H., Zhihao, L., Jing, X., ... & Rong, L. (2023). Resource utilization and development of phosphogypsum-based materials in civil engineering. Journal of Cleaner Production, 135858.

·         Jahangir, H., Soleymani, A., & Esfahani, M. R. (2022). Investigating the confining effect of steel reinforced polymer and grout composites on compressive behavior of square concrete columns. Iranian Journal of Science and Technology, Transactions of Civil Engineering, 1-17.

·         Wang, C. Q., Chen, S., Huang, D. M., Huang, Q. C., Li, X. Q., & Shui, Z. H. (2023). Safe environmentally friendly reuse of red mud modified phosphogypsum composite cementitious material. Construction and Building Materials368, 130348.

-       In the conclusion section, it is suggested to add a brief summary first, and then present the bullet conclusions.

Round 2

Reviewer 1 Report

all comments were addressed 

Author Response

Dear Reviewer,

Once again, thank you very much for your comments and hope to learn more from you.

Kind regards.

Reviewer 2 Report

The attempt to back up the added value is obvious

As per the previous comment, the testbed could benefit from superposition of the way the the measurement are being made.

Some figures in this version have been distorted and are unreadable - this needs caution.

A final proof reading of the language is suggested.

Reviewer 3 Report

The revised manuscript applied all suggestions and answered all questions.

Author Response

(The authors gave the same response as above.)
